# Podoconiosis: Clinical spectrum and microscopic presentations

**Wendemagegn Enbiale**[1,2]*, **Almut Böer-Auer**[3,4◉], **Bereket Amare**[1], **Kristien Verdonck**[5], **Gail Davey**[6,7‡], **Johan van Griensven**[5‡], **Henry J. C. de Vries**[2,8,9◉]

**1** Bahir Dar University, College of Medicine and Health sciences, Bahir Dar, Ethiopia, **2** University of Amsterdam, Department of Dermatology, Amsterdam, The Netherlands, **3** Dermatologikum Hamburg, Hamburg, Germany, **4** Department of Dermatology, Münster University, Münster, Germany, **5** Institute of Tropical Medicine, Antwerp, Belgium, **6** Brighton and Sussex Medical School, Falmer, Brighton, United Kingdom, **7** School of Public Health, Addis Ababa University, Addis Ababa, Ethiopia, **8** Center for Sexual Health, Department of Infectious Diseases, Public Health Service of Amsterdam, Amsterdam, The Netherlands, **9** Amsterdam Institute for Infection and Immunology, Infectious Diseases, Amsterdam, The Netherlands

◉ These authors contributed equally to this work.
‡ GD and JG also contributed equally to this work.
* wendemagegnenbiale@gmail.com

**Data Availability Statement:** All relevant data are within the manuscript and its Supporting Information files.

## Abstract

### Background

Podoconiosis is a skin Neglected Tropical Disease (skin NTD) that causes lymphoedema, and affects barefooted subsistence farmers in some tropical countries. The clinical presentation and histopathologic correlates of podoconiosis have been understudied. Here, we systematically document the clinical and histopathologic spectrum of podoconiosis.

### Methods

This is a cross-sectional study in Durbete, Ethiopia from February 2018 to October 2019. Dermatologists performed a patient history, physical examination, filariasis test strip, and skin biopsy for histopathologic examination. The results were summarised and a descriptive statistical analysis and Wilcoxon rank sum test with continuity correction was done.

### Results

We recruited 289 patients for the study, 178 (61.6%) had stage 1 or 2 podoconiosis, and 111(38.4%) stage 3 to 5 podoconiosis. 188 (64.1%) had a family history of podoconiosis. In 251 (86.9%) patients, both legs were affected by podoconiosis and in 38 (13.1%) only one leg was affected. 220 (77.5%) patients had warty lesions, 114 (39.4%) had nodules. The median number of episodes of Acute Dermato-Lymphangio-Adenitis (ADLA) reported by the patients in the last three months was 2 (interquartile range (IQR) 1–4). Increased episodes of ADLA were significantly associated with stage 3–5 podoconiosis (P = 0.002), while burning pain in the feet was more common in stage 1 or 2 podoconiosis. Stage 3–5 disease was histopathologically characterised by epidermal and dermal thickening, verrucous acanthosis, inflammatory cell infiltrates (predominantly lymphoplasmacytic), dilated and

**Funding:** The authors received no specific funding for this work.

**Competing interests:** The authors have declared that no competing interests exist.

ectatic and a reduced number of lymphatic vessels, eccrine ductal hyperplasia, and sclerosis such as thickened collagen bundles.

## Conclusion

We provide a detailed description of the different clinical patterns, associated clinical findings and the histopathologic spectrum of podoconiosis at different stages of the disease. Our observations should serve as a guide to classifying patients with podoconiosis for prognostic assessment and treatment decision.

Author summary

Podoconiosis is a skin Neglected Tropical Disease (skin NTD) that causes swelling of the lower extremities. The disease affects barefooted subsistence farmers in some tropical countries. It is caused by destruction of the lymphatic system in the legs, which is critical for the transportation of body fluids. Podoconiosis is physically disabling with significant psycho-social impact.

In Ethiopia alone more than 1.5 million people are affected by podoconiosis. The past 50 to 60 years generated substantial evidence on the disease distribution, genetic influence, psycho-social impact and clinical management. Yet, systematic information about the various clinical manifestations and histopathologic features of podoconiosis is sparse. We therefore recorded in 289 podoconiosis patients their history, and disease-related findings of the lower extremities. We also took blood and tissue samples for laboratory examination. In summary, this study provides a description of the different clinical manifestations and microscopic tissue findings of various podoconiosis stages from mild to advance. Our observations are a guide to classifying patients with podoconiosis, based on clinical and microscopic tissue definitions. Classification can help in patient management, therapeutic follow-up, prioritisation of resources, epidemiological surveillance and future research to improve the quality of life of patients with podoconiosis.

## Introduction

Podoconiosis is a neglected tropical disease (NTD) affecting barefooted subsistence farmers mainly in tropical countries of Africa, Southeast Asia, Central and South America. Globally the disease affects about 4 million peoples. Ethiopia is the most affected country with an estimated 1.5 million cases [1,2]. It causes chronic non-filarial lymphoedema that has a profound and negative impact on numerous health-related, psychosocial and economic aspects of the lives of affected patients and communities in endemic regions [3–5].

The aetiology of podoconiosis is not known, yet based on the geographic distribution, the affected communities, and its clinical presentation, podoconiosis is likely caused by long-term exposure to red clay soil derived from volcanic rock in rural highland areas [3].

Podoconiosis is a localised disease, primarily starting from the feet, progressively involving the legs, yet in most cases remaining below the knee. Clinically the disease severity is classified in 5 stages (S1 Table) [6]. In stages 1 and 2, simple lymphoedema management can completely reverse the clinical changes, but if the condition is not managed with appropriate foot care and consistent shoe wearing, it can lead to subsequent irreversible stages with progressive swelling

and serious disfigurement [3]. Over half of the patients report stigma, impaired mobility and daily activity, resulting in considerable morbidity and limitation of economic performance [7].

Over the past half a century many aspects of the disease epidemiology, inheritance, psycho-social impact and therapeutic management have been studied [7–13]. Yet, clinical studies on aetiology, histopathology, and the systematic documentation of the course of the disease by trained physicians are sparse. The systematic description of clinical findings and the corresponding histopathological characteristics are essential to further clarify the aetiology and pathophysiology of podoconiosis. To the best of our knowledge, to date only three studies investigated the histopathology of podoconiosis, of which the first is limited to specific morphologic descriptions while the other two are brief case reports [13–15].

Here, we examined the clinical presentation and associated complications of patients with podoconiosis, and their histopathologic correlates in the North of Ethiopia. The aim of this study is a systematic documentation of the clinical spectrum and histopathology of podoconiosis to advance patient management, therapeutic follow-up, prioritisation of resources, epidemiological surveillance and to further future research to improve the quality of life of patients with podoconiosis.

## Methods

### Ethical considerations

The study was conducted according to the principles of the Declaration of Helsinki and approval was granted from The National Research Ethics Review Committee (Ref. No. 3.10/189/2017) and Amhara Public Health Research Institute (Ref. No. A.P.H.I. T/SH/DA/01/795). Study information sheets were read to the patients in the local language (Amharic). The patients were then asked to sign a consent form written in Amharic. Patient names and identifiable information were removed, and only aggregate non-identifiable data were used for the publication. The electronic databases with patient identifiable information were kept on the password protected computer of the principal investigator.

### Study design

This is a cross-sectional study of the clinical features and histopathology of podoconiosis patients at the Durbete Podoconiosis Prevention and Treatment Centre in Durbete, Ethiopia from February 2018 to October 2019. The reporting is in line with the "Strengthening the Reporting of Observational Studies in Epidemiology" (STROBE) guidelines [16].

### General setting

Ethiopia is located in East Africa and has a population of approximately 117,697,600. The country is a federal state with nine regional and two special administrations [17], and is challenged with limited health service coverage (39% of the population) [18].

### Local setting

The Amhara regional state lies in the north west of the country with a population of 21.2 million, 85% of whom live in rural settings [19]. The region is divided into 13 administrative zones and 137 districts (called *woreda*s). This region has a podoconiosis prevalence of 3.7% and preventive shoe-wearing is strongly advised [20,21]. Durbete is a district in South Achefer zone at an altitude ranging from 1914 to 2045 meters above sea level. Podoconiosis is endemic in the district. The Durbete Podoconiosis Prevention and Treatment Centre is a specialist/referral care centre in Amhara region, West Gojam zone, South Achefer district, Durbete

town. The centre manages 400 to 600 podoconiosis patients annually and supports the local podoconiosis patient association by awareness raising, shoe distribution and community education.

## Sample size

For district population size (N): 150.000, with % frequency of outcome factor (i.e. different clinical manifestations or complications) in the population (*p*): 25%, confidence limits as % of 100 (absolute +/- %), (*d*):5%, for cluster surveys-design effect (*DEFF*):1 and 95% confidence interval ($Z^2\alpha/2$), the minimum sample size required was calculated as 288 using the sample size calculation for population frequency.

Sample size $n = [DEFF^*Np\ (1-p)]/ [(d^2/Z^2_{1-\alpha/2}{}^*(N-1) + p^*(1-p)]$

## Study population and procedure

We invited all new patients who had not previously received lymphoedema morbidity management. We recorded relevant patient history, examined the patient (with a focus on the feet and lower legs), and staged the disease according to a validated podoconiosis staging system (S1 Table) [4]. To standardise the recording, we used written case definitions for type of lymphoedema, mossy foot, ADLA and signs of fungal infection (S1 Text).

A consultant dermatologist (WE) examined all patients and made a diagnosis based on the patient history and the clinical presentation (Box 1). Four other physicians independently

---

### Box 1. Clinical criteria used for the diagnosis of podoconiosis [22]

#### Diagnostic criteria

*Major criteria*

1. Lower limb lymphoedema
2. Residing in endemic area during development of lymphoedema
3. Prolonged barefoot exposure
4. Mossy feet in slipper pattern

*Minor criteria*

1. Nodules
2. Toe fusion
3. Both feet affected
4. Positive Stemmer's sign (i.e. a test to identify dorsal foot lymphoedema. It consists of pinching and lifting a skin fold at the base of the second toe. This test is positive if the skin cannot be lifted.)
5. Burning sensation of the feet
6. Family history of podoconiosis
7. Negative filarial microscopic/antigen test

---

**Diagnostic definitions**

1. *Definitive podoconiosis*: 3 major; or 2 major plus 2 minor; or 1 major plus 5 minor criteria

2. *Probable podoconiosis*: 2 major; or 1 major plus 2 minor; or 5 minor criteria

assessed the photographic documentation of the different clinical presentations based on dermatologic morphologic descriptions [23]. For economic reasons and considering the inconvenience of invasive procedures, we performed a Filariasis Strip Test (FST), and a 6 mm tissue sample from the dorsum of foot of every fourth patient who consented to these procedures (S2 Text). In two previous studies on surgical wide excision (nodulectomy) for podoconiosis, we documented that wound healing was satisfactory, also in patients with lymphoedema. We therefore considered that the risk of complications such as wound infection, ADLA, or hyperpigmentation following 6 mm skin punch biopsy would be low [22,24].

Specimens were stained with haematoxylin and eosin (HE) and Periodic Acid-Schiff (PAS) procedures. WE and a general pathologist (BA) performed the microscopic evaluation of each slide. Discrepancies were resolved by a dermatopathologist (ABA). Information on the histopathological presentation of the epidermis, dermis, lympho-vasculature, and inflammatory cell infiltration was documented, and the presence of double breaking foreign body material was ruled out using polarised light microscopy.

## Data variables

Patient history and clinical findings (S2 Text) were recorded using a standardised questionnaire and pictures were taken. Two dermatologists independently reviewed the photographs from suspected lesions for most of the patients, and consensus was reached on the cutaneous conditions and type of lymphoedema.

## Data collection and statistical analysis

Blood was collected and Filarial antigen tests (FTS) (BinaxNOW Filariasis card test, India) were performed by a trained laboratory technician. Tissue samples for biopsy were collected either by a dermatologist or a surgeon (S2 Text). Patient history and clinical data were entered into Microsoft Excel database by a dedicated person and cross-validated by the main investigator.

We first summarised sociodemographic characteristics, medical history, and clinical findings for the entire patient population using descriptive statistics. Next, we summarised histopathological findings for the subgroup of patients with available results. Finally, we assessed the association between clinical characteristics and podoconiosis-related complications, staging (stage 1&2 versus 3–5), and histopathologic features and expressed the strength of association with odds ratios and 95% confidence intervals, and using the Wilcoxon rank sum test with continuity correction. All statistical analyses were done with R version 3.5.0 (R Foundation for Statistical Computing, Vienna, Austria) [24].

## Results

### Socio-demographic characteristics of podoconiosis patients

In total, 289 podoconiosis patients were involved in the study and 167 (57.8%) were male. The median age was 50 (IQR 35–65) years, and 85.2% of the participants had no formal education.

Only 40.1% of the patients had visited a health facility in the past for their leg swelling (without receiving treatment). For those who previously visited a health facility for their leg swelling, the median time between disease onset and care seeking was 15 (IQR 5–28) years. More than three fourths (75.4%) of the patients were farmers or shepherds (Table 1). All the 289 patients were barefoot or wore shoes inconsistently in the years before symptoms appeared. About two-thirds of patients (65.0%) had a family history of podoconiosis.

**Table 1. Sociodemographic characteristics and medical history of patients with podoconiosis in Durbete Podoconiosis Prevention and Treatment Center, Durbete, Ethiopia, February 2018 to October 2019 (n = 289).**

| Variable | Count (%)[a] |
|---|---|
| Sex | |
| Women | 122 (42.2) |
| Men | 167 (57.8) |
| Age, *years*: median (IQR) [range] | 50 (35–65) [12 – 82] |
| Schooling | |
| None | 171 (59.2) |
| Informal schooling | 75 (26.0) |
| Elementary school completed | 37 (12.8) |
| Secondary school completed | 6 (2.0) |
| Occupation | |
| Farmer | 203 (70.2) |
| Shepherd | 15 (5.2) |
| Daily labourer | 21 (7.3) |
| Student | 23 (8.0) |
| Merchant | 5 (1.7) |
| Unemployed | 9 (3.1) |
| Other (retired, civil servant, children) | 13 (4.5) |
| Age when patients got first pair of shoes, *years*: median (IQR) [range] [b] | 18 (10–27) [0–78] |
| Consistent shoe wearing[c] | |
| No | 171 (66.5) |
| Yes | 86 (33.5) |
| Family members diagnosed with podoconiosis | |
| No | 101 (33.9) |
| Yes | 188 (64.1) |
| If yes, which family member | |
| Parent | 66 (35.1) |
| Sibling | 46 (24.5) |
| Uncle or aunt | 37 (19.7) |
| Child | 28 (14.9) |
| Grandparent | 11 (5.8) |
| Sought treatment in the past | |
| No | 173 (59.9) |
| Yes | 116 (40.1) |
| Time between disease onset and seeking care, *years*: median (IQR) [range] | 15 (5–28) [0–70] |

IQR: interquartile range

[a]Except for quantitative variables, for which median, interquartile range, and range are given

[b]Missing for 9 patients

[c]Missing for 32 patients

## Clinical characteristics of podoconiosis

At the time of examination, the median time patients had leg swelling was 20 (IQR 10–32) years. There was no significant difference in disease duration between patients diagnosed with mild (stage 1 & 2) *versus* advanced (stages 3–5) podoconiosis (p-value = 0.1) and similarly, there was no association between disease duration and the type of lymphoedema (p-value = 0.2) (Fig 1).178 (61.6%) patients had mild stage podoconiosis while the rest had advanced stage.251 (86.9%) patients had both legs affected, while 38 (13.1%) had podoconiosis only in one leg. 104 (36%) patients had asymmetrical stages of the two legs and 51 (18.5%) also had asymmetry in the type of lymphoedema (one leg waterbag and the other fibrotic/sclerotic). A small number (1.4%) of patients had leg swelling extending above the knee. 220 (77.5%) patients had hyperkeratotic (mossy) lesions on one or both feet. 114 (39.4%) patients had nodules (Figs 1 and 2). The reported median number of episodes of ADLA in the last three months was 2 (IQR 1–4). About 100 (34.7%) patients had oozing foot lesions and 130 (45.1%) had foul-smelling feet. 248 (85.8%) patients had associated signs of fungal infection on their feet at the time of examination. Those with fungal infections had an increase incidence (OR = 7.3) of ADLA than those without (P<0.05). Increased episodes of ADLA were significantly associated with stage 3–5 disease (P< 0.05), while burning pain on the feet was much more common in stages 1 & 2 (Tables 2 and 3).

## Histopathology

Sixty five (31 stage 1&2, and 34 stage 3–5) tissue samples were analysed. As far as epidermal abnormalities, the vast majority of samples showed orthokeratosis/hyperkeratosis (61/65, 94%) most with compact cornification (58/65, 89%), and two-thirds with verrucous acanthosis (42/65, 65%). The epidermis was markedly hyperkeratotic with a mean thickness of 1.65mm (range 0.8–3.2 mm. Focal spongiosis was seen in 22/65 (34%) samples.

Typical dermal histopathological abnormalities were the presence of a lymphocytic cell infiltrate (in all 65 samples), located in the papillary dermis (in 56/65, 86%) and in the superficial reticular dermis (in 55/65, 86%). Apart from the lymphocytic infiltrate in all samples, 60/65 (94%) had an additional mast-cell infiltration, and 46/65 (71%) additional plasma-cell infiltration. The papillary dermis was thickened in 56/65 (86%) and showed signs of sclerosis with coarse, thickened, and vertically arranged collagen bundles in the papillary dermis, and horizontally arranged bundles in the reticular dermis. Comparing stages 1&2 versus stages 3–5 podoconiosis, we noticed more signs of sclerosis in the advanced stage samples: media-sclerosis of blood vessel walls (p<0.001), eccrine duct hyperplasia (p<0.001), dilated and ectatic lymphatic vessels (respectively 24/31, % vs 14/34, %, p<0.001) and thickening of collagen bundles (p<0.001, Table 4 and Fig 3). PAS stain for hyphae & spores, and polarisation for double refractive foreign body substances were negative in all cases.

## Discussion

We describe here the different clinical patterns, associated clinical presentations and histopathologic presentations of podoconiosis. Based on his outpatient clinic experience, EW Price in the 1980s described some of the clinical patterns, but this knowledge needed a timely update and validation by community based studies [1]. All the other recent studies on clinical presentations are based on case reports, lack photography or use inadequate terms or morphologic descriptions [2,9–11,23–26]. Given the large number (based on an initial power calculation) and representative distribution of participants, we feel confident that the clinical and histopathological findings presented here truly reflect the podoconiosis diseases spectrum.

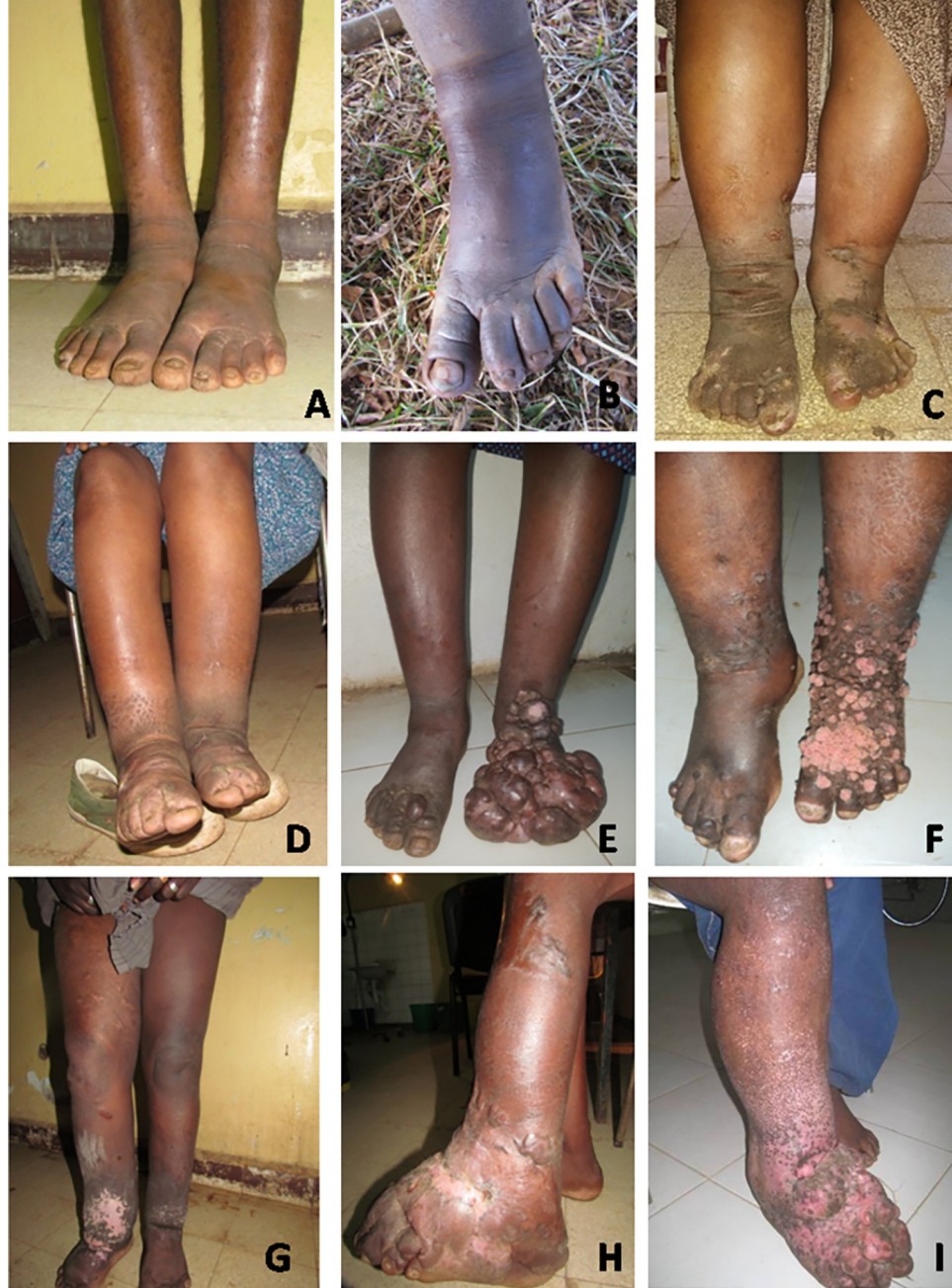

**Fig 1. Most common characteristics of the various stages of podoconiosisin, in Durbete Podoconiosis Prevention and Treatment Centre, Durbete, Ethiopia, February 2018 to October 2019.** A) Stage 1: overnight reversible oedema, with increased skin marking and hyperkeratosis on the base of the toes of both the right and left foot. B) Stage 1: splaying of the right forefoot, accentuation of the skin markings on the metatarsophalangeal joint. C) Stage 2: bilateral lymphoedema below the knee. Area of scarring is seen on the lower leg and foot (from traditional bloodletting). Hyperpigmented, warty hyperkeratotic papules covering the dorsum of the feet extending to the ankle. Nodules on the right forefoot. D) Stage 2: Pitting lymphoedema below the knee, with dry skin. E) Stage 2(right foot) and stage 3 (left foot). Right foot: pitting lymphoedema on the shin, few papules and nodules on the dorsum anterior one third of the foot and toes. Left foot: smooth surface with multiple nodules and tumorous masses covering the whole foot and flexural ankle with extension to the anterior distal one third of the lower leg. The foot is totally deformed and the toes are no longer visible. Based on the extension of the nodules the right foot is stage 2 (nodules below the ankle only) and the left foot stage 3 (nodules extending above the ankle). F) Stage 3: Bilateral lymphoedema. On the right leg the swelling is pitting on the proximal half of the lower leg and non-pitting fibrotic edema on the distal one third of the lower leg and foot. Scattered, infiltrative papules on the dorsal distal one third of the foot and toes. Hyperkeratotic and

depigmented rough papules and nodules covering the distal one third of the left lower leg, and dorsum of the foot. G) Stage 4 (right) and stage 2 (left). Right foot: non pitting oedematous swelling of the leg extending above the knee with an area of skin depigmentation around the ankle. Left foot: non pitting swelling of the leg below the knee. H) Stage 5: fibrotic globular swelling of the left foot and ankle with ankylosis of the ankle joint, multiple areas of scarring (from traditional bloodletting). I) Stage 5: right leg with nodules and rubbery to woody hard tumors on the dorsum of the foot with band like redundant skin on flexural ankle (pillowy oedema) and ankle fixation.

Nodules can occur from stage 2 onwards and 39.4% of our patients had nodules on the feet or lower legs. The programmatic implication of nodules on the foot is that most of those patients require surgical intervention [13,27,28]. In previous publications, oedema in podoconiosis was considered not to extend above the knee [12,28,29]. However, in our study 4 (1.4%) patients had swelling above the knee (stage 4 disease). Documentation from the 1990s reported that 2.4% of podoconiosis patients had above-knee lymphoedema and other recent studies, reported 5.9%-18% of patients with stage 4 podoconiosis [9,10,23].

In most studies, the standard description of podoconiosis is that it results in "progressive bilateral swelling of the lower legs" and most studies that have documented stages of podoconiosis have failed to explore on the evidence of the variation/symmetry of involvement [1,2,8,10]. This study has showed that asymmetry in leg involvement is common as far as the type of lymphoedema and hyperkeratosis between both legs involved. These findings suggest that even though podoconiosis is linked to genetic factors, the type of lymphoedema manifested is unlikely to be genetically linked. The finding also poses a question of why such significant asymmetric manifestations should have an environmental cause, if both feet are exposed equally to the same environment.

Price stated that most patients presented late [1]. The late presentation in our study also suggests that the structural and individual barriers preventing patients seeking care need to be identified and addressed. Stigma could be an important factor in late presentations. Like many other studies [30–32], we found a large proportion of patients to have a family history of podoconiosis. Family history of similar illness is an important factor in differentiating podoconiosis from lymphatic filariasis.

In our study, 77.5% of patients had hyperkeratotic papules in a 'slipper' distribution on the feet. Other studies reported a wide variation (38.5 to 97.95) in the clinical presentation [9,10,33,34]. From our clinical observation, hyperkeratotic papules are common in podoconiosis with no previous treatment history and among those who do not wear protective shoes consistently. The wide variation in prevalence seen across the different studies may be explained by the absence of a standard clinical definition and/or the limited clinical experience of data collectors.

Distinguishing the characteristics by disease stage can help the clinical management of podoconiosis, such as the choice of treatment option, and follow-up after treatment. Moreover, clear disease staging criteria are critical to further our understanding of podoconiosis pathophysiology, compile evidence-based prevention measures and as endpoints in the evaluation of innovative therapies. The existing podoconiosis staging criteria has been used since 2008 [6]. Considering the findings in this article, we recommend revision of the clinical staging. In the new staging criteria in addition to clarification/rephrasing the existing clinical stages we highlighted the necessary criteria for each stage, removed the redundancy and included additional clinical findings (S1 Table).

As to the prevalence of ulcers, similar inconsistencies with earlier literature were found. In our study 9.7% of the patients had an associated leg ulcer. One study in West Ethiopia documented 37.7% of patients examined to have associated ulcers [10], but another study in the same population reported 7.2% of patients to have leg ulcers [32]. An epidemiological study in Northern Ethiopia reported that 53% of patients had an ulcer [9].

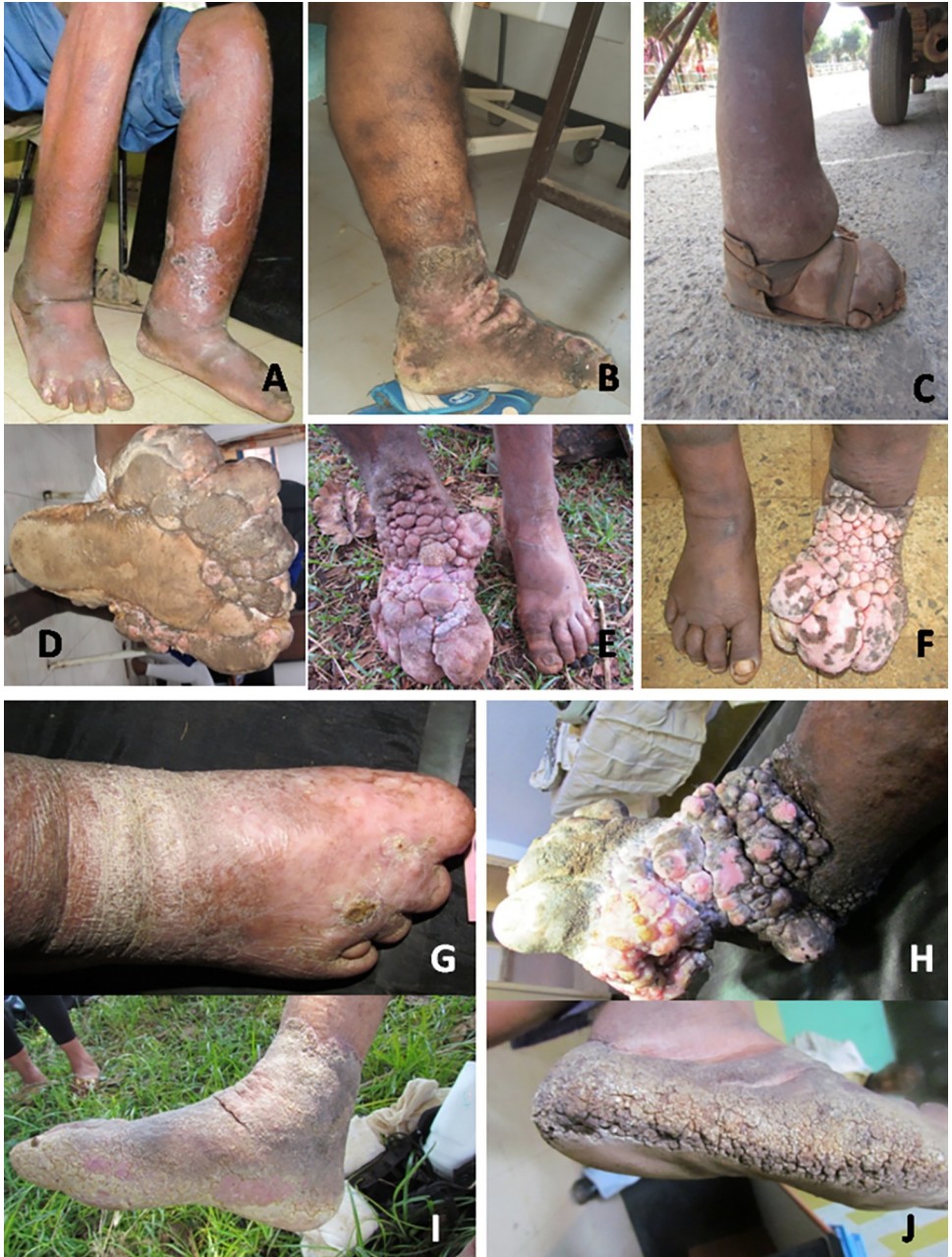

**Fig 2. Clinical variations and complications of podoconiosis, in Durbete Podoconiosis Prevention and Treatment Centre, Durbete, Ethiopia, February 2018 to October 2019.** A) Stage 2 podoconiosis patient with Acute Dermato-Lymph-Angitis (ADLA) presenting with pitting oedema of the left leg with epidermal exfoliation on the distal half of the lower leg. B) Stage 3 fibrotic oedema of the left leg below the knee with sclerotic hyperpigmentation and hypopigmentation accentuated on the shin and a fibrotic ridge on the flexural ankle. C) Stage 2: Water-bag type oedema with pitting and soft swelling and a flask-like appearance with the neck around the knee and wider base on the ankle and oedematous foot with a smooth and dumpy surface around the shoe strap areas. D) Stage 3 podoconiosis, multiple nodules with plantar foot involvement. E) Asymmetric podoconiosis with right leg stage 3 diseases with fibrotic depigmented nodules on foot and above the ankle, while the left leg is not affected. F) Asymmetric lymphoedema with the right leg in stage1, and the left leg in stage 3 podoconiosis with depigmented nodules and toe fusion. G) Stage 5: oedematous right foot with scarring of the distal half of the dorsum of the foot, bone resorption of all the toes. H) Stage 3 podoconiosis with multiple coalescent nodules on the dorsum of the foot extending to the sole of the foot and above the ankle with maceration and oozing. I) Fibrotic lymphoedema of the right leg with a continuous sock-like yellowish crust covering the whole foot and lower leg, and band like skin invagination on the flexural ankle joint. J) Hyperpigmented warty papules coalescing and covering the lateral feet and cracking in moccasin-like configuration.

**Table 2. Clinical features of patients with podoconiosis in Durbete Podoconiosis Prevention and Treatment Center, Durbete, Ethiopia, February 2018 to October 2019 (n = 289).**

| Variable | Count (%)[a] |
|---|---|
| Burning feeling or pain at the time of the examination | |
| No | 236 (81.7) |
| Yes | 53 (18.3) |
| Stage[b] | |
| Stage1 | 24 (8.3) |
| Stage 2 | 154 (53.3) |
| Stage 3 | 102 (35.3) |
| Stage 4 | 4 (1.4) |
| Stage 5 | 5 (1.7) |
| Symmetry | |
| Both feet same stage | 185 (64.0) |
| Both feet different stage | 66 (22.8) |
| Podoconiosis on one foot only | 38 (13.2) |
| Mossy hyperkeratotic papillomata | |
| No | 69 (23.9) |
| On one extremity | 59 (20.4) |
| On both extremities | 161 (55.7) |
| Presence of nodules | |
| No | 175 (60.6) |
| On one extremity | 49 (17.0) |
| On both extremities | 65 (22.4) |
| Type of lymphoedema | |
| Water bag type on both extremities | 165 (59.8) |
| Fibrotic type on both extremities | 60 (21.7) |
| One extremity waterbag, other extremity fibrotic type | 51 (18.5) |
| Number of ADLA episodes in past three months: median (IQR) [range][b] | 2 (1–3) [0 – 6] |
| Presence of ulcers | |
| No | 261 (90.3) |
| Yes | 28 (9.7) |
| Presence of fungal infection | |
| No | 41 (14.2) |
| Yes | 248 (85.8) |
| Presence of toe fusion | |
| No | 231 (79.9) |
| On one extremity | 14 (4.8) |
| On both extremities | 44 (15.3) |
| Presence of foul smell | |
| No | 159 (50.0) |
| Yes | 130 (45.0) |

ADLA: acute dermato-lymphango-adenitis; IQR: interquartile range

[a]Except for quantitative variables, for which median, interquartile range, and range are given

[b] Based on the side with the most advanced lesions

The most disabling problem associated with podoconiosis is ADLA, which needs emergency medical intervention and prevention strategies. About 86% of the cases in our study had at least one episode of ADLA during the three-month period preceding the date of

**Table 3. Associated morbidity of stage 1–2 versus stage 3–5 podoconiosis in Durbete Podoconiosis Prevention and Treatment Centre, Durbete, Ethiopia, February 2018 to October 2019.**

|  | Stage 1–2 podoconiosis (n = 178) | Stage 3–5 podoconiosis (n = 111) | Odds ratio (95% confidence interval) | p-value[a] |
|---|---|---|---|---|
|  | Median (IQR) | Median (IQR) |  |  |
| ADLA episodes [b] | 2 (1–3) | 3 (1–4) | 1.3 (1.1–1.5) | 0.002 |
| Time off work due to ADLA[b] | 4 (3–4) | 4 (3–4) | 1.2 (1.0–1.4) | 0.07 |
|  | Count (%) | Count (%) |  |  |
| Fever | 10 (5.6) | 24 (21.6) | 4.6 (2.2–10.6) | <0.0001 |
| Burning feeling or pain[c] | 43 (24.4) | 10 (9.2) | 0.3 (0.1–0.6) | 0.002 |
| Presence of eczema[c] | 29 (16.3) | 9 (8.2) | 0.5 (0.2–1.0) | 0.07 |
| Presence of ulcers[c] | 16 (9.0) | 12 (10.9) | 1.2 (0.6–2.7) | 0.7 |
| Fungal infection[c] | 151 (84.8) | 96 (87.3) | 1.2 (0.6–2.5) | 0.7 |

ADLA: acute dermato-lymphangitis; IQR: interquartile range

[a]Wilcoxon rank sum test with continuity correction orPearson's Chi-squared test with Yates' continuity correction

[b]Three months prior to examination

[c]Affecting one or both lower extremities

examination. ADLA frequency was significantly higher in those with stage 3–5 podoconiosis and those with associated interdigital fungal infection. The frequency of ADLA varies in studies from 5 to 23.3 episodes per year [9–11] and interdigital lesions were found to be the main risk factor [34]. One factor explaining this wide variation in reported episodes may be recall bias. We tried to minimise this by asking the history over the last 3 months instead of one year. ADLA frequency data would be much more accurate if the study was prospective. More importantly, in almost all previous studies which have documented ADLA, an inadequate case definition (a reddish, hot, swollen leg with a painful groin) has been used, or the case definition is not clear [9–11]. In our study comprising patients with skin of colour, exfoliative dermatitis was the dominant clinical finding. Whereas most studies use erythema/red skin as hallmark sign for ADLA, this is not helpful in the dermatologic evaluations of individuals with a skin of colour [35]. The reported frequency of ADLA episodes in lymphoedema patients with podoconiosis is much more frequent than that of lymphoedema secondary to lymphatic filariasis [36] suggesting the need for public health programs to improve access to prevention and case management.

Histopathologic presentations of podoconiosis have a lot of similarity to those seen in long-standing filarial lymphoedema in regard to epidermal and dermal thickening, epidermal verrucous acanthosis and sclerosis [15,37–38]. However, in podoconiosis, infiltrates consisted predominantly of mast cells, plasma cells and lymphoplasmacytic cells, while lymphocytes predominate the infiltrates in filariasis and mast cells have not been emphasised in this condition [37]. Our previous study on the histology of nodules from podoconiosis patients demonstrated similar findings [13]. Mast cells are commonly seen in fibrotic skin disorders, and it has been hypothesised that they may be involved in the pathogenesis of fibrosis [39]. Podoconiosis also differs from filariasis in terms of vascular abnormalities. While filariasis shows dilated and tortuous lymphatics, in stages 3-5podoconiosis, the lymphatics are reduced and blood vessels are dilated, increased and often show a sclerotic wall [38–42]. In stage 1–2 podoconiosis, the cellular infiltrates are much more intense, probably showing a more active inflammatory process. Compared to the advanced stages of podoconiosis, in stages 1–2, the lymphatic vessels were more numerous and dilated, with indication of collateral formation. The reduction of lymphatic vessels, sclerotic blood and lymphatic vessel walls, eccrine ductal hyperplasia,

**Table 4. Histopathologic findings and associations in skin biopsies from 65 patients with stage 1–2 versus stage 3–5 podoconiosis, in Durbete Podoconiosis Prevention and Treatment Centre, Durbete, Ethiopia, February 2018 to October 2019.**

| Clinical feature | Stage 1–2 podoconiosis (n = 31) | Stage 3–5 podoconiosis (n = 34) | All patients (n = 65) | p-value[a] |
|---|---|---|---|---|
| | Count (%) | Count (%) | Count (%) | |
| Ortho/hyperkeratosis | 28 (90) | 33 (97) | 61 (94) | |
| Focal | 0 (0) | 0. (0) | 0 (0) | |
| Confluent | 28 (90) | 33 (97) | 61 (94) | 0.9 |
| Compact | 27 (87) | 31 (91) | 58 (89) | 0.9 |
| Basketweave | 0 (0) | 1 (3) | 1 (2) | 1.0 |
| Parakeratosis | 1 (3) | 0 (0) | 1 (2) | 1.0 |
| focal | 1 (3) | 0 (0) | 1 (2) | 1.0 |
| confluent | 0 (0) | 0 (0) | 0 (0) | - |
| Hypergranulosis | 16 (52) | 11 (32) | 27 (42) | 0.2 |
| Spongiosis[b] | 9 (29) | 13 (38) | 22 (34) | 0.6 |
| Acanthosis | 23 (74) | 19 (56) | 42 (65) | 0.2 |
| regular | 17 (55) | 15 (44) | 32 (49) | |
| irregular | 6 (19) | 2 (6) | 8 (12) | |
| verrucous | 6 (19) | 4 (12) | 10 (15) | |
| Papillomatosis | 3 (10) | 2 (6) | 5 (8) | 0.9 |
| Edema of papillary dermis | 6 (19) | 12 (35) | 18 (28) | 0.2 |
| Infiltrate | 31 (100) | 34 (100) | 65 (100) | |
| in dermal papillae | 27 (87) | 29 (85) | 56 (86) | 0.9 |
| in superficial reticular dermis | 29 (94) | 26 (79)[b] | 55 (86) | 0.1 |
| in deep reticular dermis | 22 (71) | 21 (62) | 43 (66) | 0.6 |
| in subcutis | 4 (13) | 6 (18) | 10 (15) | 0.9 |
| perivascular | 28 (90) | 33 (97) | 61 (94) | 0.5 |
| interstitial | 9 (29) | 3 (9) | 12 (18) | 0.08 |
| perivascular and interstitial | 9 (29) | 3 (9) | 12 (18) | 0.08 |
| Eccrine ductal changes | | | | |
| >2 layers of ductal cells | 4 (13) | 23(68) | 27 (42) | 0.001 |
| reticulate proliferation | 4 (13) | 12 (35) | 16 (25) | 0.07 |
| miliaria | 2 (6) | 6 (18) | 8 (12) | 0.3 |
| Type of infiltrate | | | | |
| lymphocytic | 31 (100) | 34 (100) | 65 (100) | - |
| lymphoeosinophilic | 0 (0) | 0 (0) | 0 (0) | - |
| lymphoplasmacytic | 23 (74) | 23 (68) | 46 (71) | 0.8 |
| mast cells | 29 (94) | 31 (94)[b] | 60 (94) | 0.9 |
| lymphohistiocytic | 15 (48) | 14 (41) | 29 (45) | 0.7 |
| neutrophilic | 0 (0) | 1 (3) | 1 (2) | 1.0 |
| Blood vessels | | | | |
| increase | 20 (65) | 27 (79) | 47 (72) | 0.3 |
| decrease | 1 (3) | 0 (0) | 1 (2) | 1.0 |
| dilation / ectasia | 26 (84) | 22 (65) | 48 (74) | 0.1 |
| sclerosis of vessel walls | 1 (3) | 16 (47) | 17 (26) | 0.001 |
| Lymphatic vessels | | | | |
| increase | 15 (48) | 9 (26) | 24 (37) | 0.1 |
| decrease | 0 (0) | 7 (21) | 7 (11) | 0.02 |
| dilation / ectasia | 24 (77) | 14 (41) | 38 (58) | 0.001 |
| Collagen | | | | |

*(Continued)*

**Table 4.** (Continued)

| Clinical feature | Stage 1–2 podoconiosis (n = 31) | Stage 3–5 podoconiosis (n = 34) | All patients (n = 65) | p-value[a] |
|---|---|---|---|---|
| | Count (%) | Count (%) | Count (%) | |
| Thickened bundles | 17 (55) | 29 (85) | 46 (71) | 0.01 |
| Sclerosis of connective tissue | 5 (16) | 17 (50) | 22 (34) | 0.008 |

[a]Wilcoxon rank sum test

[b]All patients with spongiosis had mild forms

thickening of collagen and sclerotic connective tissue are signs of a fully developed stage of the disease.

The strengths of this study are that each patient was evaluated by an expert dermatologist, and that the description of the clinical pattern was made based on the morphology. The clinical findings of a large number of mostly untreated patients with different stages of the disease were systematically documented and revision of the staging criteria was recommended. This is also the first study into the histopathologic substrate of the various podoconiosis stages, allowing a better understanding of the spectrum of clinical presentations and to correlate it with associated histopathology, morbidities and severity.

One limitation of this study is that patients were recruited from the treatment centre and hence only give a picture of podoconiosis patients seeking health care, and may not represent those who do not have access to or knowledge about the existence of such service. As information on the clinical history was collected retrospectively by patient interview, there may be some recall bias especially on the duration of the disease or the date of the first symptoms. The newly suggested clinical staging also needs to be field tested.

The findings of this study emphasise the need for standardisation of clinical presentations of podoconiosis, both in clinical and research settings. Second, the detailed morphologic description of the disease with histopathological correlates is expected to serve as a reference for clinicians for diagnosis and management. This information will also be important for public health in budgeting, resource allocation, and monitoring, but also in the design of awareness and prevention campaigns.

## Conclusion

The clinical features of podoconiosis include bilateral but mostly asymmetrical leg lymphoedema. Clinically, symptoms of stages 1 & 2 podoconiosis are a burning sensation or/and itching of the forefoot, splaying of the forefoot, plantar oedema, hyperkeratosis and increased skin markings. Later, debilitating bilateral lymphoedema of the lower leg occurs with or without skin changes such as hyperkeratosis, nodules and 'mossy' papillomatosis. Accompanying oedema may be either soft or hard and fibrotic. Two out of five patients have multiple smooth surface or hyperkeratotic nodules or tumours dominantly on the dorsum of the foot extending to the ankle area. Other clinical variants are unilateral involvement, toe fusion, toe resorption, maceration, and ulcerative podoconiosis. The histopathologic changes demonstrate broad similarities with the lymphoedematous stage of filariasis but also have some differences in the vascular abnormalities and the type of infiltrates.

We suggest that further research includes the establishment of prospective cohorts, population based epidemiological studies, field testing of the revised staging and more in-depth histological studies to further the understanding of the pathogenesis of podoconiosis, including potential infectious associations.

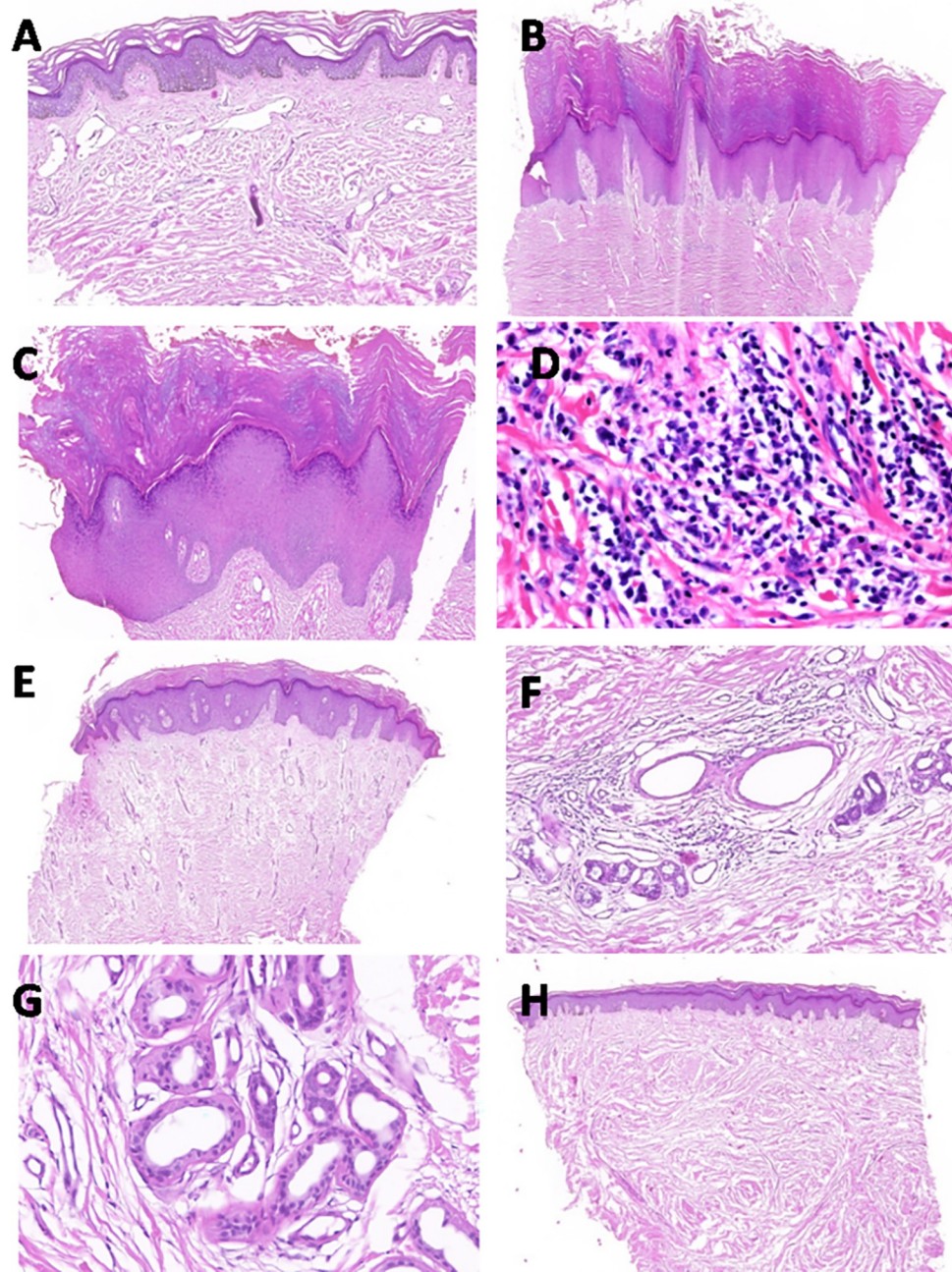

**Fig 3. Histopathologic characteristics of podoconiosis in skin biopsies from 65 patients with stages 1 & 2 versus stages 3–5 podoconiosis, in Durbete Podoconiosis Prevention and Treatment Centre, Durbete, Ethiopia, February 2018 to October 2019.** A) Stages 1 podoconiosis: Basket woven hyperkeratosis, acanthosis, and ectatic and dilated lymphatic vessels (40X magnification). B) Stages 2 podoconiosis: Compact hyperkeratosis and acanthosis with elongated rete ridges, sclerosis with numerous linear, perpendicular arranged capillaries in the papillary dermis (20X magnification). C) Stages 3 podoconiosis: Verrucous hyperplasia of the epidermis (40X magnification). D) Stage 3 and 4 podoconiosis: Dense dermal infiltrate with lymphocyte, histocyte and plasma cells (100X magnification). E) Stage 3 podoconiosis: Compact hyperkeratosis and acanthotic epidermis with increased dermal vasculature (20X magnification). F) Stage 4 podoconiosis: Sclerosis vessel with perivascular infiltrate (100X magnification). G) Stage 4 podoconiosis: Dilated eccrine gland with area of sclerosis around the gland (100X magnification). H) Stage 5 podoconiosis: Sclerosis dermis (both papillary and reticular) with loss of adenexal structure and vasculatures (20X magnification).

## Supporting information

**S1 Table. Annex A, Clinical staging's.**
(DOCX)

**S1 Text. Annex B, Operational definitions.**
(DOCX)

**S2 Text. Annex C, Procedures and variables.**
(DOCX)

## Acknowledgments

We thank all the study participants in Durbete for their willingness to participate in this study. We also acknowledge senior staff and residents (Dr. Asressie Mamo, Dr. Debas Tesfa, Dr. Wosen Ketema, and Dr. Yared Getachew) of the dermatovenerology department in Bahir Dar University College of Medicine and Health Sciences for their contribution on the review and description of patient's clinical presentation photographs.

## Author Contributions

**Conceptualization:** Wendemagegn Enbiale.

**Data curation:** Wendemagegn Enbiale, Kristien Verdonck.

**Formal analysis:** Kristien Verdonck.

**Funding acquisition:** Henry J. C. de Vries.

**Investigation:** Wendemagegn Enbiale, Almut Böer-Auer, Bereket Amare.

**Methodology:** Wendemagegn Enbiale.

**Project administration:** Wendemagegn Enbiale.

**Resources:** Almut Böer-Auer, Henry J. C. de Vries.

**Supervision:** Almut Böer-Auer, Henry J. C. de Vries.

**Validation:** Almut Böer-Auer, Gail Davey, Johan van Griensven, Henry J. C. de Vries.

**Visualization:** Wendemagegn Enbiale, Almut Böer-Auer, Kristien Verdonck, Gail Davey, Johan van Griensven, Henry J. C. de Vries.

**Writing – original draft:** Wendemagegn Enbiale.

**Writing – review & editing:** Wendemagegn Enbiale, Almut Böer-Auer, Bereket Amare, Kristien Verdonck, Gail Davey, Johan van Griensven, Henry J. C. de Vries.

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
