## [Decision Letter · Decision Letter 0]

8 Mar 2022

Dear Dr Enbiale,

Thank you very much for submitting your manuscript "Podoconiosis: Clinical spectrum and microscopic presentations" for consideration at PLOS Neglected Tropical Diseases. As with all papers reviewed by the journal, your manuscript was reviewed by members of the editorial board and by several independent reviewers. The reviewers appreciated the attention to an important topic. Based on the reviews, we are likely to accept this manuscript for publication, providing that you modify the manuscript according to the review recommendations. 

Please not comments from each of the reviewers. I suggest emphasising the reference below which illustrates how well patients heal in your set up if you don't have the information for this cohort.

Surgical nodulectomies can heal in patients with lymphoedema secondary to podoconiosis in resource-poor settings

British Journal of Dermatology

2017-10 | Journal article

DOI: 10.1111/bjd.15420

Part of ISSN: 0007-0963

Show less detail

Language

English

URL

http://dx.doi.org/10.1111/bjd.15420

Added

2020-08-26

Last modified

2020-08-26

Source: Wendemagegn Enbiale

Sincerely,

Claire Fuller

Guest Editor

Fabiano Oliveira

Deputy Editor

Please not comments from each of the reviewers. I suggest emphasising the reference below which illustrates how well patients heal in your set up if you don't have the information for this cohort.

Surgical nodulectomies can heal in patients with lymphoedema secondary to podoconiosis in resource-poor settings

British Journal of Dermatology

2017-10 | Journal article

DOI: 10.1111/bjd.15420

Part of ISSN: 0007-0963

Show less detail

Language

English

URL

http://dx.doi.org/10.1111/bjd.15420

Added

2020-08-26

Last modified

2020-08-26

Source: Wendemagegn Enbiale

Reviewer's Responses to Questions

**Key Review Criteria Required for Acceptance?**

**Methods**

-Are the objectives of the study clearly articulated with a clear testable hypothesis stated?

-Is the study design appropriate to address the stated objectives?

-Is the population clearly described and appropriate for the hypothesis being tested?

-Is the sample size sufficient to ensure adequate power to address the hypothesis being tested?

-Were correct statistical analysis used to support conclusions?

-Are there concerns about ethical or regulatory requirements being met?

Reviewer #1: Noting the assessment of participants consulting health care before the date of the thier current presentation, might you consider including the estimate of how many consulted traditional healers for example?

From personal observation noting evidence of traditional healers surigcally puncturing the lymphoedematous limb which has potential to increase risk of ADLA and leaves behind hyperpigmentation, this hasnt been commented on. I assume this is not noted in your cohort? Might be worth mentioning.

Also worth in the introduction stressing your previous study showing the safety and satisfacvtory healing of surgery in this patient group encouraing you that biopsy was appropriate and reasonable to do for research purposes in this setting. ALso might be worth stating the healing and complication rate of your biopsies if can just to reassure/educate the lymphoedema community who otherwise would avoid trauma.

Reviewer #2: The objectives of the study are well described with a clear testable hypothesis stated.

The study design is appropriate. 

The population of study is clearly described; however I miss information about Durbete town, e.g., its altitude above sea level, which is important for soil characteristics and thus, it poses a risk for the development of podoconiosis. 

The sample size is sufficient. 

Statistical analysis was used to support conclusions.

There are no concerns about ethical or regulatory requirements.

**Results**

-Does the analysis presented match the analysis plan?

-Are the results clearly and completely presented?

-Are the figures (Tables, Images) of sufficient quality for clarity?

Reviewer #1: The title of table 2: I wonder if this might be beter "clinical features distributed by stage" rather than "risk factors"?

Reviewer #2: The analysis is complete and fully describe; it does match the analysis plan.

The results are clearly and completely presented. However, I find some discrepancies along the text regarding data about clinical descriptions. 

- In the abstract (line 37) the main conclusion expressed was the reduction in lymphatic vessels, however in results (lines 287) the main findings were dilated and ecstatic lymphatic vessels, also shown in table 4. 

- In my opinion, the description of the authors about the asymmetry of clinical manifestations of the disease may be confusing. Patients may have uni or bilateral affection; however, in the abstract it is said that 251 had podoconiosis in both legs and 104 patients had only one leg affected, while in lines 201-203 the authors describe 251 patients with both legs affected, and 38 with one leg podoconiosis. 

- Lines 87-90: there is another paper that describes the histopathology of two patients with podoconiosis: Med Clin (Barc). 2015 Nov 20;145(10):446-51. doi: 10.1016/j.medcli.2014.12.020. PMID: 25726310.

Figures are clear and precise.

**Conclusions**

-Are the conclusions supported by the data presented?

-Are the limitations of analysis clearly described?

-Do the authors discuss how these data can be helpful to advance our understanding of the topic under study?

-Is public health relevance addressed?

Reviewer #1: (No Response)

Reviewer #2: The conclusions are well supported by the data presented

I do not find significant limitations of analyisis.

The authors discuss how their results can be helpful to advance in the understanding of the disease

Public health relevance is addressed.

**Editorial and Data Presentation Modifications?**

Reviewer #1: There seems to be a formatting issue on my pdf version with many of the words lacking spaces between them. It may be a soft ware glitch.

Reviewer #2: There are numerous typographical errors, essentially lack of spaces.

Line 198 versus should not be italicized.

**Summary and General Comments**

Reviewer #1: This is an extensive and ccomprehenisve clinicopathological study, the most extensive of its kind ever performed providing robust insights into the clinico-pathological features of podoconiosis as well as supporting earlier assumptions of the risk factors for the quality of life altering aspects of this disorder.

It is also proposing a modification and update to the previous pragmatic version of the staging scheme which will enhance subsequent field sutides.

Reviewer #2: The study is interesting and solid. I wonder about the evolution of patients after the skin biopsy, since any invasive procedure -however subtle- may pose a risk for infection, specially in this kind of environment. I believe it would be interesting to add whether there were clinical complications (e.g. infection) or not.

PLOS authors have the option to publish the peer review history of their article (what does this mean?). If published, this will include your full peer review and any attached files.

Reviewer #1: Yes: Dr L Claire Fuller

Reviewer #2: No

Figure Files:

Data Requirements:

Reproducibility:

References

---

## [Decision Letter · Decision Letter 1]

18 Apr 2022

Dear Dr Enbiale,

We are pleased to inform you that your manuscript 'Podoconiosis: Clinical spectrum and microscopic presentations' has been provisionally accepted for publication in PLOS Neglected Tropical Diseases.

Best regards,

Claire Fuller

Guest Editor

Fabiano Oliveira

Deputy Editor

<style type="text/css">p.p1 {margin: 0.0px 0.0px 0.0px 0.0px; line-height: 16.0px; font: 14.0px Arial; color: #323333; -webkit-text-stroke: #323333}span.s1 {font-kerning: none

</style>

Reviewer's Responses to Questions

**Key Review Criteria Required for Acceptance?**

**Methods**

-Are the objectives of the study clearly articulated with a clear testable hypothesis stated?

-Is the study design appropriate to address the stated objectives?

-Is the population clearly described and appropriate for the hypothesis being tested?

-Is the sample size sufficient to ensure adequate power to address the hypothesis being tested?

-Were correct statistical analysis used to support conclusions?

-Are there concerns about ethical or regulatory requirements being met?

Reviewer #2: Everything is clear now

**Results**

-Does the analysis presented match the analysis plan?

-Are the results clearly and completely presented?

-Are the figures (Tables, Images) of sufficient quality for clarity?

Reviewer #2: The analysis match the analysis plan and the results are clear and completely presented.

**Conclusions**

-Are the conclusions supported by the data presented?

-Are the limitations of analysis clearly described?

-Do the authors discuss how these data can be helpful to advance our understanding of the topic under study?

-Is public health relevance addressed?

Reviewer #2: The conclusions are supported by the data presented.

**Editorial and Data Presentation Modifications?**

Reviewer #2: (No Response)

**Summary and General Comments**

Reviewer #2: (No Response)

PLOS authors have the option to publish the peer review history of their article (what does this mean?). If published, this will include your full peer review and any attached files.

Reviewer #2: No

---

## [Editor Report · Acceptance letter]

18 May 2022

Dear Dr Enbiale,

We are delighted to inform you that your manuscript, "Podoconiosis: Clinical spectrum and microscopic presentations," has been formally accepted for publication in PLOS Neglected Tropical Diseases.

Best regards,

Shaden Kamhawi

co-Editor-in-Chief

Paul Brindley

co-Editor-in-Chief
